# Impact of the COVID-19 pandemic on physical therapy practice in Saudi Arabia

Walaa Elsayed[1]*, Faisal Albagmi[1], Mehwish Hussain[2], Mohammed Alghamdi[1], Ahmed Farrag[3]

1 Department of Physical Therapy, College of Applied Medical Sciences, Imam Abdulrahman Bin Faisal University, Dammam, Saudi Arabia, 2 Department of Public Health, College of Public Health, Imam Abdulrahman Bin Faisal University, Dammam, Saudi Arabia, 3 Basic Science Department, Faculty of Physical Therapy, Cairo University, Cairo, Egypt

* whelsayed@iau.edu.sa

**Data Availability Statement:** All relevant data are within the paper and its Supporting Information files. Further, the data underlying the results presented in the study are available from: https://

## Abstract

### Purpose

The new Coronavirus (COVID-19) pandemic has caused significant impact on the medical sector worldwide, including physical therapy (PT). The purpose of this study was to investigate the impact of the COVID-19 pandemic on the PT services, and the associated psychological distress endured by PT practitioners in Saudi Arabia.

### Methods

A cross-sectional study was conducted to survey on-duty PT practitioners using a web-based questionnaire. Licensed PT practitioners working in Saudi Arabia (n = 265) participated and completed all the survey questions. The questionnaire comprised 30 questions covering the sociodemographic data and the outcome measures, which included the impact of the pandemic on the PT practice, use of telerehabilitation, administrative response during the pandemic, and PT practitioners' anxiety measured by the General Anxiety Disorder-7 scale.

### Results

During the lockdown, disruptive impact on the PT practice was reported by most of the participants (80%). The majority of PT clinics/departments were either partially (43.8%) or completely (31.3%) shutdown, and therapists treated patients less than usual. Around 30% of participants reported using a telerehabilitation approach during the pandemic to communicate with patients, and about 33% received online courses and webinars to adapt the PT practice in response to the pandemic. About 21% of participants endured moderate to severe levels of anxiety, which was more pronounced among females.

### Conclusion

The COVID-19 pandemic significantly impacted the PT services in Saudi Arabia. Consequently, the number of patients treated was reduced, and therapists suffered notable psychological stress. Furthermore, although implemented, adaptive administrative measures

datadryad.org/stash/share/_JjcrdXB993D2n7D8M
cOfIrhAKU59U4yQsT87SAfJ5Q.

**Funding:** The authors received no specific funding
for this work.

**Competing interests:** The authors have declared
that no competing interests exist.

were inadequate. Physical therapy practitioners and administrative authorities are encouraged to prioritize training and implementation of telerehabilitation as a likely prospective approach of PT practice.

## Introduction

The new coronavirus (COVID-19) emerged first in China in December 2019. Since then, the number of patients with COVID-19 has been dramatically increasing all over the globe [1]. Since the COVID-19 pandemic was declared by the World Health Organization (WHO) in March, 2020, preventive measures were implemented worldwide to help control its spread [2, 3]. Different measures including lockdown, social distancing, travel limitations, and rearrangement of health administrations were mandated globally. To diminish the risk of transmitting the infection to both the public and medical staff, health care providers either postponed patients' visits or used digital-based consultation whenever possible [4–6]. From their side, many patients chose to maintain social distancing and decided to defer their medical visits. Accordingly, The COVID-19 pandemic has clearly impacted the delivery of medical care.

Physical Therapy (PT) services have an integral role in the management of COVID-19 cases, either during illness or after recovery [7–9]. The evolving recommendations of physical distancing during the pandemic forced physical therapists to defer patient's appointments or to use the recommended personal protection equipment while dealing with patients. Accordingly, the use of telerehabilitation was recommended as an alternative service-providing method during the pandemic [10].

The influence of the COVID-19 pandemic on medical practice has recently been studied in several healthcare disciplines. The impact varied among different health specialties. It was reported that 52% and 27.8% of spine surgeons were moderately or severely negatively impacted by the pandemic, respectively [11]. In India, orthopedic and PT services administration was subjected to delay due to the pandemic [12]. However, despite of the global recommendation to implement telemedicine, its utilization was restricted due to the need for physical examination and treatment of patients to ensure efficient delivery of services [12]. In Portugal, 54% of PT services were adapted from direct contact to distant monitoring [13], and around 73% of physiotherapists interrupted their practice during the pandemic [4]. In countries with a large number of COVID-19 patients, orthopedic services like elective surgeries were paused. Accordingly, PT became a primary healthcare service to ensure the quality of care during the pandemic [14]. Despite ranked second of the Eastern Mediterranean countries with the highest total count of COVID-19 cases [15], little is known about how the medical services in Saudi Arabia were impacted by the COVID-19 pandemic. A single-institution retrospective study reported a reduction of the admitted pediatric oncology patients since the start of the pandemic [16].

The psychological impact of disease outbreaks on healthcare workers has been documented in the literature. Previous studies reported an adverse psychological impact including anxiety and depression on frontline healthcare workers [17–19]. Nevertheless, the circumstances associated with the COVID-19 pandemic, including the lockdown and other precautionary measures, are unprecedented. The COVID-19 pandemic imposed a significant change to the social, economic, and healthcare systems. A recent study reported that the percentage of staff with severe anxiety was equal between the high- and low-risk healthcare workers involved in the care of COVID-19 patients [20]. Fear of contracting the infection is a psychological burden

that applies to all staff. Additionally, the psychological burden due to staff reorganization and close contact with a COVID-19 patient during rehabilitation represents a challenge unique to PT practitioners [18, 20].

To the best of our knowledge, none of the previous studies examined the impact of the COVID-19 pandemic on the PT practice and the associated practitioners' anxiety in Saudi Arabia. Shedding light on the method of service modification from a PT perspective during the pandemic, will help identify recommendations for advancing physical therapy practice in Saudi Arabia. Moreover, investors will focus on the needed areas of training to enhance the healthcare service. Therefore, this study aimed to investigate the impact of the COVID-19 pandemic on the PT profession in Saudi Arabia in terms of PT practice disruption, service adaptation and the involvement of therapists in administrative service planning, use of telerehabilitation and forms of education received, and the practitioners' psychological stress.

## Methods

### Design

A cross-sectional design was implemented to survey the physical therapy practitioners in Saudi Arabia. A web-based questionnaire was used to investigate the impact of the COVID-19 pandemic on the PT practice, and anxiety. The ethical approval was obtained from the Institutional Review Board of the Prince Sultan Military College of Health Sciences (IRB-2020-PT-032).

### Sampling

Physical therapists working in Saudi Arabia were invited through emails and social media to fill a web-based questionnaire. The sample size was calculated using the Raosoft calculator and based on an estimated population size of 5000 licensed physical therapists according to data posted on the World Physiotherapy website [21], with a confidence level of 90%, and *p*-value of 0.05. Accordingly, a sample of at least 257 therapists was targeted. Inclusive criteria were licensed physical therapy practitioners including physical therapists, and PT assistants working inside Saudi Arabia only. Participants gave their consent through clicking a link on the online survey.

### Instrument

A web-based survey composed of 30 questions was used for this study. It covered the participant's demographic data and the following domains: impact of the pandemic on PT practice, use of telerehabilitation, receiving education to adapt PT services, involvement in the planning of PT services, and psychological stress. The survey tool was constructed by the authors in accordance with the guidelines of the world physiotherapy on adapting PT services during the COVID-19 Pandemic [9] and based on a modified published questionnaire applied on spine surgeons [11], and another study that described PT practice during the pandemic in Portugal [4]. The impact of the pandemic on PT practice was tested by a developed set of twelve questions. Six questions with four possible answers each were used to quantify the impact of the pandemic on PT services. The four answers had an ascending order, and each was given a score between 1 and 4. The average score of the six questions was calculated to quantify the impact on PT services. A threshold of 2 was considered to classify impact. A score of less than 2 denoted negligible impact, while a score of 2 or more denoted disruptive impact. The rest of the questions were used to describe how the service was modified during the pandemic. The psychological stress was assessed using the General Anxiety Disorder-7 (GAD-7) scale [22].

## Procedures

Content validity of the questionnaire items were first determined by consensus among the authors. Then, the questionnaire was pilot tested to identify potential ambiguities. Ten respondents completed the web-based survey and were asked to write their feedback about the questions. Then, the respondents were interviewed and requested to provide responses to the survey items that were rephrased using alternative wording. Their responses were checked against what they provided in the web-based questionnaire. Afterwards, the questionnaire was amended accordingly based on the pilot study results, and the formal questionnaire was constructed using QuestionPro software and disseminated during September 2020 through the following channels: emails, SMS, WhatsApp, LinkedIn, Twitter, Instagram, Facebook, and Snapchat.

## Data analysis

Data were analyzed using SPSS version 26 (SPSS, Chicago, IL). Completed responses for each variable were examined. Descriptive statistics were performed for all the variables in the form of frequency (percentages) distribution. Median with interquartile range was used for the GAD-7 score. Univariable analyses were performed to examine the significant differences between the defined parameters among different demographic characteristics. Chi-square test was used for the categorical parameters. Mann-Whitney U test was used to compare the GAD-7 score between age-groups, gender, years of experience, and type of employers. Kruskal Wallis test was used to compare the GAD-7 score among different work regions. The significance level ($\alpha$) was set at $P \leq 0.05$. Variables with significant univariable effects were further processed using multivariable analysis. Logistic regression was performed to estimate the association between the demographic characteristics and study parameters. Odds ratio (OR) are presented with a 95% confidence interval (CI).

## Results

### Demographic characteristics

A total of 424 physical therapy practitioners participated, of whom 265 (62.5%) participants completed all the survey questions. The majority of participants were Saudi therapists (92.7%), who were mostly females (61.9%), aged less than 30 years (67.5%), with five years or less of experience (63.1%), and working in governmental institutions (60.9%). They were primarily ranked as junior (48%) or senior (35.3%) therapists, while consultants represented 4.5%. Only 17.6% had a higher degree (master or doctoral degree).

### Impact on physiotherapists' practice

During the lockdown, most of the PT clinics were either partially (43.8%) or completely (31.3%) shutdown, and only 6.4% counted solely on telerehabilitation. After the lockdown was lifted, only 13.1% of the clinics continued complete shutdown, while more than half of the clinics were partially open. During the pandemic, 55.6% of participants reported managing fewer patients than usual. Most of the physiotherapists (>80%) encountered a disruptive impact on their practice (Table 1). There was no significant association between demographic characteristics and the quantified impact on PT practice.

### Use of telerehabilitation

about 31.7% (n = 112) of participants reported using a telerehabilitation approach during the pandemic. Phone calls were the commonest approach followed by messages and emails, while

**Table 1. Impact of the COVID-19 pandemic on PT practice.**

| | | Impact | | |
| --- | --- | --- | --- | --- |
| | | **Negligible** | **Disruptive** | **P-Value** |
| **Age (years)** | **Till 30** | 27 (14.1%) | 165 (85.9%) | 0.340 |
| | **Above 30** | 19 (18.3%) | 85 (81.7%) | |
| **Gender** | **Male** | 20 (15.7%) | 107 (84.3%) | 0.915 |
| | **Female** | 26 (15.3%) | 144 (84.7%) | |
| **Work Region** | **Eastern** | 21 (18.4%) | 93 (81.6%) | 0.518 |
| | **Central** | 13 (15.9%) | 69 (84.1%) | |
| | **Western** | 7 (12.7%) | 48 (87.3%) | |
| | **Northern** | 3 (20.0%) | 12 (80.0%) | |
| | **Southern** | 2 (6.5%) | 29 (93.5%) | |
| **Years of experience** | **≤5** | 26 (14.5%) | 153 (85.5%) | 0.572 |
| | **>5** | 20 (16.9%) | 98 (83.1%) | |
| **Type of Employer** | **Governmental institution** | 30 (15.8%) | 160 (84.2%) | 0.848 |
| | **Non-Governmental institution** | 16 (15.0%) | 91 (85.0%) | |

video conferences and software applications were the least used. Significant associations were found between the use of telerehabilitation and gender, years of experience, and type of employer (Table 2). Males used the telerehabilitation more than females (OR: 1.7, 95% CI: 1.1–2.7). The greater the experience the more the likelihood of using tele-rehabilitation (OR: 2.1, 95% CI: 1.3–3.3). The use of telerehabilitation was more common in the governmental organizations (OR: 1.6, 95% CI: 1.01–2.6). Videoconferencing was more commonly used among males ($P = 0.003$) and experienced physiotherapists ($P = 0.029$). Older ($P = 0.049$) and more experienced ($P = 0.003$) physiotherapists reported greater usage of messages and emails.

**Table 2. Response to the COVID-19 pandemic.**

| | | Use of telerehabilitation approach | | | | Education received | | | |
| --- | --- | --- | --- | --- | --- | --- | --- | --- | --- |
| | | Did you use tele-rehabilitation approach during the COVID-19 pandemic? | | | | Have you received education to adapt your PT practice during the COVID-19 pandemic? | | | |
| | | **Yes** | **No** | **P-Value** | **OR (95% CI)** | **Yes** | **No** | **P-Value** | **OR (95% CI)** |
| **Age (years)** | **Till 30** | 67 (28.5%) | 168 (71.5%) | 0.084 | | 57 (29.7%) | 135 (70.3%) | 0.299 | |
| | **Above 30** | 44 (37.6%) | 73 (62.4%) | | | 37 (35.6%) | 67 (64.4%) | | |
| **Gender** | **Male** | **54 (38.8%)** | **85 (61.2%)** | **0.021** | 1.7 (1.1–2.7) | **50 (39.4%)** | **77 (60.6%)** | **0.018** | 1.8 (1.1–3.0) |
| | **Female** | **58 (27.1%)** | **156 (72.9%)** | | 1 | **45 (26.5%)** | **125 (73.5%)** | | 1 |
| **Work Region** | **Eastern** | 36 (26.3%) | 101 (73.7%) | 0.15 | | 33 (28.9%) | 81 (71.1%) | 0.783 | |
| | **Central** | 29 (31.2%) | 64 (68.8%) | | | 29 (35.4%) | 53 (64.6%) | | |
| | **Western** | 21 (32.8%) | 43 (67.2%) | | | 16 (29.1%) | 39 (70.9%) | | |
| | **Northern** | 7 (36.8%) | 12 (63.2%) | | | 6 (40.0%) | 9 (60.0%) | | |
| | **Southern** | 19 (47.5%) | 21 (52.5%) | | | 11 (35.5%) | 20 (64.5%) | | |
| **Years of experience** | **< = 5** | **57 (25.7%)** | **165 (74.3%)** | **<0.001** | 1 | **48 (26.8%)** | **131 (73.2%)** | **0.019** | 1 |
| | **>5** | **55 (42.0%)** | **76 (58.0%)** | | 2.1 (1.3–3.3) | **47 (39.8%)** | **71 (60.2%)** | | 1.8 (1.1–3.0) |
| **Type of Employer** | **Governmental institution** | **78 (35.6%)** | **141 (64.4%)** | **0.045** | 1.6 (1.0–2.6) | 67 (35.3%) | 123 (64.7%) | 0.107 | |
| | **Non-Governmental institution** | **34 (25.4%)** | **100 (74.6%)** | | 1 | 28 (26.2%) | 79 (73.8%) | | |

Bold numbers indicate significant results at $P ≤ 0.05$. OR: odds ratio, CI: confidence interval

## Education received

Only 95 (32.9%) participants received education to adapt the PT practice during the COVID-19 pandemic. Online seminars (24.1%) was the commonest form of education. Gender and experience were significantly associated with education (Table 2). Males ($P = 0.018$) and experienced ($P = 0.019$) physiotherapists received education to adapt the PT practice almost as twice (OR: 1.8, 95% CI: 1.1–2.95) as their female and inexperienced counterparts. Team meetings and in-person education were more attended by males ($P = 0.015$), while educational emails were more utilized by experienced physiotherapists ($P = 0.009$) working in governmental institutions ($P = 0.006$).

## Administrative response to the pandemic

Around 65% (n = 175) of physiotherapists reported that during the COVID-19 pandemic, they were not involved in the strategic and operational planning of service delivery in their clinics. However, almost 50% (n = 130) reported that their clinic responded appropriately and adapted the PT services according to the guidelines and recommendations issued by scientific organizations. Involvement in service planning and the clinic's response were similarly influenced by participants' age ($P = 0.004$ and $P = 0.015$, respectively), gender ($P = 0.03$ and $P = 0.002$, respectively), and experience ($P<0.001$). Older and more experienced male therapists were more involved in the planning of service delivery in their PT department (Table 3).

During the COVID-19 pandemic, 60.2% of the therapists reported managing patients with acute more than chronic conditions. Experience ($P = 0.029$) and type of employer ($P<0.001$)

**Table 3. Involvement of physiotherapists in service planning and their clinic response.**

| | | During the COVID-19 pandemic, have you been involved in the planning of service delivery? | | | | During the COVID-19 pandemic, did your clinic respond appropriately and adapted the PT services according to the guidelines? | | | | |
| --- | --- | --- | --- | --- | --- | --- | --- | --- | --- | --- |
| | | Yes | No | P- Value | OR (95% CI) | Yes | No | Not Sure | P- Value | OR (95% CI) |
| Age (years) | Till 30 | 47 (27.5%) | 124 (72.5%) | **0.004** | 1 | 47 (27.5%) | 124 (72.5%) | 51 | **0.015** | 1 |
| | Above 30 | 42 (45.2%) | 51 (54.8%) | | 2.2 (1.3–3.7) | 42 (45.2%) | 51 (54.8%) | 24 | | 2.0 (1.2–3.3) |
| Gender: | Male | 46 (41.1%) | 66 (58.9%) | **0.03** | 1.8 (1.1–3.0) | 46 (41.1%) | 66 (58.9%) | 26 | **0.002** | 2.4 (1.5–3.9) |
| | Female | 43 (28.3%) | 109 (71.7%) | | 1 | 43 (28.3%) | 109 (71.7%) | 49 | | 1 |
| Work Region | Eastern | 38 (38.0%) | 62 (62.0%) | 0.109 | | 38 (38.0%) | 62 (62.0%) | 33 | 0.245 | |
| | Central | 20 (26.7%) | 55 (73.3%) | | | 20 (26.7%) | 55 (73.3%) | 20 | | |
| | Western | 12 (25.0%) | 36 (75.0%) | | | 12 (25.0%) | 36 (75.0%) | 15 | | |
| | Northern | 8 (53.3%) | 7 (46.7%) | | | 8 (53.3%) | 7 (46.7%) | 1 | | |
| | Southern | 11 (42.3%) | 15 (57.7%) | | | 11 (42.3%) | 15 (57.7%) | 6 | | |
| Years of experience | < = 5 | 38 (24.2%) | 119 (75.8%) | **<0.001** | 1 | 38 (24.2%) | 119 (75.8%) | 47 | **<0.001** | 1 |
| | >5 | 51 (47.7%) | 56 (52.3%) | | 2.9 (1.7–4.8) | 51 (47.7%) | 56 (52.3%) | 28 | | 2.3 (1.4–3.9) |
| Type of Employer | Governmental institution | 63 (36.4%) | 110 (63.6%) | 0.2 | | 63 (36.4%) | 110 (63.6%) | 44 | 0.124 | |
| | Non-Governmental institution | 26 (28.6%) | 65 (71.4%) | | | 26 (28.6%) | 65 (71.4%) | 31 | | |

Bold numbers indicate significant results at $P \leq 0.05$. OR: odds ratio, CI: confidence interval

Table 4. Management of different patients by physiotherapist.

| | | During the COVID-19 pandemic, which category of patients did you manage more? | | |
| --- | --- | --- | --- | --- |
| | | Acute cases | Chronic cases | P-Value |
| Age (years) | Till 30 | 96 (56.1%) | 75 (43.9%) | 0.066 |
| | Above 30 | 63 (67.7%) | 30 (32.3%) | |
| Gender: | Male | 62 (55.4%) | 50 (44.6%) | 0.165 |
| | Female | 97 (63.8%) | 55 (36.2%) | |
| Work Region | Eastern | 59 (59.0%) | 41 (41.0%) | 0.823 |
| | Central | 47 (62.7%) | 28 (37.3%) | |
| | Western | 30 (62.5%) | 18 (37.5%) | |
| | Northern | 7 (46.7%) | 8 (53.3%) | |
| | Southern | 16 (61.5%) | 10 (38.5%) | |
| Years of experience | < = 5 | **86 (54.8%)** | **71 (45.2%)** | **0.028** |
| | >5 | **73 (68.2%)** | **34 (31.8%)** | |
| Type of Employer | Governmental institution | **120 (69.4%)** | **53 (30.6%)** | **<0.001** |
| | Non-Governmental institution | **39 (42.9%)** | **52 (57.1%)** | |

Bold numbers indicate significant results at $P \leq 0.05$.

were significantly associated with the patient's conditions mostly managed during the pandemic (Table 4). Experienced therapists working in governmental institutions managed acute cases more than chronic ones. In contrast, less experienced therapists working in non-governmental institutions managed more chronic cases.

## Anxiety level

Results showed that 21.6%, 30.7%, and 47.7% of the therapists endured moderate to severe, mild, and minimal levels of anxiety during the pandemic, respectively. Physiotherapists who were younger, females, less experienced and from non-governmental institutions suffered more severe anxiety (Fig 1). The GAD-7 scores showed that female physiotherapists were significantly more anxious than their male counterparts ($P = 0.023$).

## Discussion

In the current study, the results revealed that the majority of PT practitioners (>80%) in Saudi Arabia have encountered a disruptive impact on their practice due to the pandemic. This finding could be attributed to the shutdown of the PT clinics during the pandemic. Consequently, the majority of PT practitioners managed patients less than the usual rate. Due to the shutdown and the significant drop in the patient's rate managed regularly, it is believed that financial drawbacks affecting primarily the private sector has occurred. Accordingly, junior physiotherapists working in the private sector were likely more financially affected by the pandemic, which can be explained by the cutoff of the salaries during the closure of many private clinics that dramatically affected the income. Our findings were consistent with previous studies showing the disruptive impact of the COVID-19 pandemic on physical therapists [4, 12], orthopedic clinics [12, 14], and other medical practices [11].

It was also shown that during the pandemic, rehabilitation services were delivered more to patients with acute conditions than chronic ones. This is mostly attributable to the shutdown of PT clinics, because patients with chronic conditions mostly receive rehabilitation services in outpatient clinics. Meanwhile, patients with acute symptoms such as COVID-19 patients with acute respiratory distress were the primary target for healthcare providers. Furthermore, the

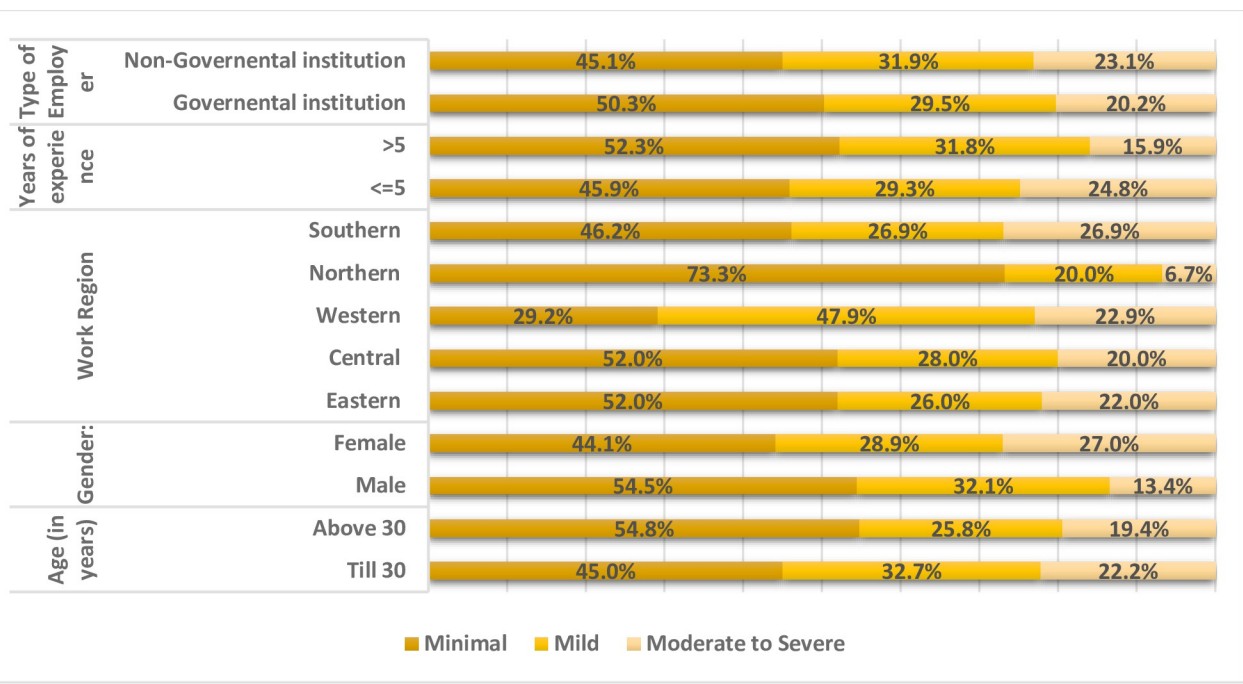

**Fig 1. Anxiety level endured by physical therapists.**

use of telerehabilitation as an alternative solution to follow up chronic patients was used only by few physiotherapists (6.4%) during the lockdown. Such response denotes a limited implementation of telerehabilitation among physical therapists.

Physical therapy constitutes an essential healthcare service in Saudi Arabia. Continuation of PT service is mandatory during such a sudden global pandemic. Almost one-third of physiotherapists reported receiving education to modify their practice. Online seminars was the most commonly used method used by physiotherapists, whilst team meetings were the least. Although the percentage of physiotherapists who reported receiving appropriate education is relatively low, the use online seminars as the primary method of education indicates that physiotherapists responded well and were committed to social distancing to minimize the risk of infection. Our findings suggest that being a male and senior physiotherapist could be a predictor for receiving education. Such unexpected finding should be alarming to the PT community and administrative authorities that educational efforts should equally be directed to the all physiotherapists regardless of gender or experience. This should be emphasized considering that >60% of the participants were female and less experienced physiotherapists, and females represent 49% of the PT working force in Saudi Arabia [21].

Almost one-third of participants used digital communication approaches to remotely provide PT services and monitor their patients through the crisis. This indicates that both the physical therapists and patients have relatively accepted and adopted digital communication methods because of the pandemic. Interestingly, remote monitoring and follow-up were mostly achieved using phone calls, while other communication methods were significantly less utilized. It seems that communication over the phone was the convenient choice for both the therapists and patients, despite its limitations compared to other methods. The use of video conferencing or software applications would be advantageous in that they provide visual and illustrative interaction that could not be achieved through phone calls [4]. Another important finding is that therapists working in the governmental sector mostly utilized digital

communication approach more than the private sector. It seems that the governmental sector may have responded promptly and adopted the telerehabilitation approach earlier than the private sector. Thus, it is recommended that the use of video conferencing as an efficient method of telerehabilitation be emphasized by the non-governmental sector to facilitate remote patient monitoring [4].

The PT practitioners need to be involved in the planning process of the PT services during exceptional circumstances such as the COVID-19 pandemic. This ensures staff understanding and compliance with the imposed measures to achieve efficient delivery of services. Our results indicate that the majority of therapists (65%) were not involved in strategic planning of services during the pandemic; and those involved in planning were primarily older and more experienced male therapists, which is acceptable as older and experienced employees are typically assigned to higher managerial positions. Around 50% of participants thought that their clinic performed well during the pandemic and planned service delivery as per the guidelines. This may reflect the impression therapists generally had regarding how their institution managed the crisis, despite not being involved in managerial planning and the significant disruptive impact they experienced because of the pandemic.

The present results showed that gender was a factor for developing severe anxiety during the COVID-19 pandemic. Physical therapists who reported enduring severe anxiety during the pandemic were females (27%) more than males (13.4%). This is consistent with previous studies showing that women are more vulnerable to developing psychological distress in both the general population and healthcare workers during a disease outbreak [20, 23]. We also found that a smaller percentage of senior physical therapists (15.9%) reported having severe anxiety compared with junior therapists (24.8%). This could be attributed to being more accustomed to the relevant occupational hazards and being able to develop resistance to stressful working conditions. These findings are in accordance with previous studies demonstrating that junior nurses are more likely to experience adverse psychological impacts during a highly infectious outbreak than their senior counterparts [24]. Furthermore, our findings indicate that younger physical therapists working in the private sector suffered slightly more severe anxiety compared with those working in the public sector. The economic strain that the pandemic has put on the private practice due to closure and disruption of services may have contributed to this. These findings suggest that further research on the economic impact of the COVID-19 pandemic on small to medium-sized privately-owned PT clinics is warranted. The current study findings would help to enhance the rehabilitation services after the abrupt changes encountered by the pandemic. Attention should be given to train physiotherapists specially juniors and keeping them scientifically updated. Alternative ways like digital PT should be implemented to help keep quality of life for the chronic conditions. Investors could benefit from this study by targeting the needed areas to advance the rehabilitation services like offering educational courses for the digital rehabilitation, psychological, or professional support seminars.

## Study limitations

the current study has limitations that should be acknowledged. The five geographical regions of the Saudi Arabia were not equally represented, which may limit the generalizability of the obtained findings. However, successfully recruiting participants more than the required minimal sample size could help provide credible data. This study was not conducted at the time of the lockdown during the peak of COVID-19 pandemic. Therefore, the data obtained were collected retrospectively. Thus, it may likely suffer the recall bias inherent with retrospective self-reported data.

## Conclusion

The COVID-19 pandemic inflicted disruptive drawbacks on the PT practice in Saudi Arabia. Accordingly, psychological distress impacted PT practitioners variably. PT services prioritized acute cases over chronic ones. The global pandemic prompted physical therapists to adapt to it variably by receiving education, using digital PT approaches, and modifying the service. The study highlighted the need for the private sector and junior staff to emphasize adaptation of services during such pandemic. Future studies should be directed to identify the competencies required to practice digital PT to facilitate effective delivery of care during crisis.

## Supporting information

**S1 File. Questionnaire.**
(PDF)

## Author Contributions

**Conceptualization:** Walaa Elsayed, Faisal Albagmi, Ahmed Farrag.

**Data curation:** Mehwish Hussain, Mohammed Alghamdi.

**Formal analysis:** Mehwish Hussain, Mohammed Alghamdi.

**Investigation:** Walaa Elsayed, Mehwish Hussain, Mohammed Alghamdi, Ahmed Farrag.

**Methodology:** Walaa Elsayed, Faisal Albagmi, Ahmed Farrag.

**Project administration:** Walaa Elsayed, Mohammed Alghamdi.

**Software:** Mehwish Hussain, Mohammed Alghamdi.

**Supervision:** Walaa Elsayed, Ahmed Farrag.

**Validation:** Mehwish Hussain.

**Visualization:** Faisal Albagmi.

**Writing – original draft:** Walaa Elsayed, Faisal Albagmi.

**Writing – review & editing:** Walaa Elsayed, Ahmed Farrag.

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
