## [Decision Letter · Decision Letter 0]

21 Feb 2022

PONE-D-21-28813Impact of the COVID-19 Pandemic on Physical Therapy PracticePLOS ONE

Dear Dr. Elsayed,

Thank you for submitting your manuscript to PLOS ONE. After careful consideration, we feel that it has merit but does not fully meet PLOS ONE’s publication criteria as it currently stands. Therefore, we invite you to submit a revised version of the manuscript that addresses the points raised during the review process.

We look forward to receiving your revised manuscript.

Kind regards,

Fahad Jibran, Ph.D.

Academic Editor

PLOS ONE

https://journals.plos.org/plosone/s/fileid=ba62/PLOSOne_formatting_sample_title_authors_affiliations.pdf".

2. Please include additional information regarding the survey used in the study and ensure that you have provided sufficient details that others could replicate the analyses.

3. During your revisions, please note that a simple title correction is required: please include the country where the study has been performed, i.e. 'Impact of the COVID-19 Pandemic on Physical Therapy Practice in Saudi Arabia'. Please ensure this is updated in the manuscript file and the online submission information.

Additional Editor Comments:

Abstract

Line 36: "about 33% received education to adapt the PT practice"

You may use online courses, webinars instead

Sampling

- Line 109: (Physical therapists working in Saudi Arabia were invited to fill a…) how they were invited?

There are many factors that adversely affect psychological state and lead to anxiety:

- Did you consider whether the participant are native Saudi Arabian or from other nationalities?

- Lockdown, travel limitations, being affected with COVID-19 or losing beloved persons affect the psychological state. Did the authors considered these factors?

Reviewers' comments:

Reviewer's Responses to Questions

**Comments to the Author**

1. Is the manuscript technically sound, and do the data support the conclusions?

Reviewer #1: Yes

Reviewer #2: Yes

2. Has the statistical analysis been performed appropriately and rigorously? 

Reviewer #1: Yes

Reviewer #2: Yes

3. Have the authors made all data underlying the findings in their manuscript fully available?

Reviewer #1: Yes

Reviewer #2: Yes

4. Is the manuscript presented in an intelligible fashion and written in standard English?

Reviewer #1: Yes

Reviewer #2: Yes

5. Review Comments to the Author

Reviewer #1: the data submitted support the conclusion drawn from study. appropriate statistical approach has been adapted. the language of paper is understandable and does not contain errors. ethical approval was done for this study.

Reviewer #2: Manuscript Number: ''PONE-D-21-28813'' entitled "Impact of the COVID-19 Pandemic on Physical Therapy Practice"

Overall, the idea of research is very interesting. I appreciate your hard work on this interesting topic. I think the work is very nice and very detailed. However, there are some comments and suggestions.

Revision Suggestions

Title

- Brief and concise

Abstract

Line 36: "about 33% received education to adapt the PT practice"

You may use online courses, webinars instead

Introduction

- Well structured

Materials and methods

- Well structured

Sampling

- Line 109: (Physical therapists working in Saudi Arabia were invited to fill a…) how they were invited?

There are many factors that adversely affect psychological state and lead to anxiety:

- Did you consider whether the participant are native Saudi Arabian or from other nationalities?

- Lockdown, travel limitations, being affected with COVID-19 or losing beloved persons affect the psychological state. Did the authors considered these factors?

Statistical analysis:

- Well structured

Results: Satisfying

Discussion: Satisfying

6. PLOS authors have the option to publish the peer review history of their article (what does this mean?). If published, this will include your full peer review and any attached files.

Reviewer #1: No

Reviewer #2: No

---

## [Author Response · Author response to Decision Letter 0]

1 Mar 2022

Response to specific reviewers and editor comments:

The authors would like to thank the editor and the reviewers for their time and their comments on our manuscript. Our response to the comments is posted beneath each original comment.

Title: please include the country where the study has been performed, i.e. 'Impact of the COVID-19 Pandemic on Physical Therapy Practice in Saudi Arabia'.

Response: The comment is addressed. The title has been updated.

Reviewer #1: the data submitted support the conclusion drawn from study. appropriate statistical approach has been adapted. the language of paper is understandable and does not contain errors. ethical approval was done for this study.

Response: the authors appreciate the reviewer’s comment.

Reviewer #2: Manuscript Number: ''PONE-D-21-28813'' entitled "Impact of the COVID-19 Pandemic on Physical Therapy Practice"

Overall, the idea of research is very interesting. I appreciate your hard work on this interesting topic. I think the work is very nice and very detailed. However, there are some comments and suggestions.

Revision Suggestions

Title

- Brief and concise

Abstract

Line 36: "about 33% received education to adapt the PT practice"

You may use online courses, webinars instead

Response: the authors appreciate the reviewer’s comment. The comment is addressed

Introduction

- Well structured

Materials and methods

- Well structured

Sampling

- Line 109: (Physical therapists working in Saudi Arabia were invited to fill a…) how they were invited?

Response: the comment is addressed. The authors clarified this information at the revised manuscript version. “Physical therapists working in Saudi Arabia were invited through emails and social media to fill a web-based questionnaire”.

There are many factors that adversely affect psychological state and lead to anxiety:

- Did you consider whether the participant are native Saudi Arabian or from other nationalities?

Response: thank you for the meticulous observation and comment. We considered this information on the demographic data collected. We found that 92.7% of the respondents were Saudi. 

- Lockdown, travel limitations, being affected with COVID-19 or losing beloved persons affect the psychological state. Did the authors considered these factors?

Response: we didn’t consider that in our survey since we used the GAD-7 score for testing anxiety which measures anxiety with no reflection for the causes behind it.

Statistical analysis:

- Well structured

Results: Satisfying

Discussion: Satisfying

---

## [Editor Report · Decision Letter 1]

6 May 2022

PONE-D-21-28813R1

Impact of the COVID-19 Pandemic on Physical Therapy Practice in Saudi Arabia

PLOS ONE

Dear Dr. Elsayed,

Thank you for submitting your manuscript to PLOS ONE. After careful consideration, we have decided that your manuscript does not meet our criteria for publication and must therefore be rejected.

Specifically:

The last date to submit revised manuscript was 07th April 2022 and today is 5th May 2022, Hence you have passed given deadline and on this account your paper is being rejected.

I am sorry that we cannot be more positive on this occasion, but hope that you appreciate the reasons for this decision.

Kind regards,

Fahad Jibran, Ph.D.

Academic Editor

PLOS ONE

Additional Editor Comments (if provided):

The last date to submit revised manuscript was 07th April 2022 and today is 5th May 2022, Hence you have passed given deadline and on this account your paper is being rejected.
---

## [Author Response · Author response to Decision Letter 1]

14 Jul 2022

Response to Reviewers’

The authors would like to thank the editor and the reviewers for their time and their comments on our manuscript. Our response to the comments is posted beneath each original comment.

Title: please include the country where the study has been performed, i.e. 'Impact of the COVID-19 Pandemic on Physical Therapy Practice in Saudi Arabia'.

Response: The comment is addressed. The title has been updated.

Reviewer #1: the data submitted support the conclusion drawn from study. appropriate statistical approach has been adapted. the language of paper is understandable and does not contain errors. ethical approval was done for this study.

Response: the authors appreciate the reviewer’s comment.

Reviewer #2: Manuscript Number: ''PONE-D-21-28813'' entitled "Impact of the COVID-19 Pandemic on Physical Therapy Practice"

Overall, the idea of research is very interesting. I appreciate your hard work on this interesting topic. I think the work is very nice and very detailed. However, there are some comments and suggestions.

Revision Suggestions

Title

- Brief and concise

Abstract

Line 36: "about 33% received education to adapt the PT practice"

You may use online courses, webinars instead

Response: the authors appreciate the reviewer’s comment. The comment is addressed. Please check line 34

Introduction

- Well structured

Materials and methods

- Well structured

Sampling

- Line 109: (Physical therapists working in Saudi Arabia were invited to fill a…) how they were invited?

Response: the comment is addressed. The authors clarified this information at the revised manuscript version on line 107. “Physical therapists working in Saudi Arabia were invited through emails and social media to fill a web-based questionnaire”.

There are many factors that adversely affect psychological state and lead to anxiety:

- Did you consider whether the participant are native Saudi Arabian or from other nationalities?

Response: thank you for the meticulous observation and comment. We considered this information on the demographic data collected. We found that 92.7% of the respondents were Saudi. 

- Lockdown, travel limitations, being affected with COVID-19 or losing beloved persons affect the psychological state. Did the authors considered these factors?

Response: we didn’t consider that in our survey since we used the GAD-7 score for testing anxiety which measures anxiety with no reflection for the causes behind it.

Statistical analysis:

- Well structured

Response: Thank you for your support

Results: Satisfying

Response: Thank you for your comment

Discussion: Satisfying

Response: Thank you for your comment

---

## [Editor Report · Decision Letter 2]

23 Nov 2022

Impact of the COVID-19 Pandemic on Physical Therapy Practice in Saudi Arabia

PONE-D-21-28813R2

Dear Dr. Elsayed,

We’re pleased to inform you that your manuscript has been judged scientifically suitable for publication and will be formally accepted for publication once it meets all outstanding technical requirements.

Kind regards,

Fahad Jibran, Ph.D.

Academic Editor

PLOS ONE
---

## [Editor Report · Acceptance letter]

1 Dec 2022

PONE-D-21-28813R2 

Impact of the COVID-19 Pandemic on Physical Therapy Practice in Saudi Arabia 

Dear Dr. Elsayed:

I'm pleased to inform you that your manuscript has been deemed suitable for publication in PLOS ONE. Congratulations! Your manuscript is now with our production department. 

Kind regards, 

on behalf of

Dr. Fahad Jibran 

Academic Editor

PLOS ONE